# Quantitative Model Study of the Psychological Recovery Benefit of Landscape Environment Based on Eye Movement Tracking Technology

**Xinhui Fei [1], Yanqin Zhang [1], Deyi Kong [1], Qitang Huang [1], Minhua Wang [1,2] and Jianwen Dong [1,2,\*]**

[1] College of Landscape Architecture, Fujian Agriculture and Forestry University, 15 Shangxiadian Rd., Fuzhou 350002, China; epsfxh@163.com (X.F.)

[2] Engineering Research Center for Forest Park of National Forestry and Grassland Administration, Fuzhou 350002, China

**\*** Correspondence: fjdjw@fafu.edu.cn; Tel.: +86-133-1377-6679

**Abstract:** From the perspective of landscape and human health, we use the Self-Rating Restoration Scale (SRRS) as a tool to explore the mental health restoration benefits brought by a landscape environment to individuals and explore the characteristics of individual movement behavior when viewing the landscape through the eye movement tracking technology. We selected average blink duration, average gaze length, average saccade amplitude, blink number, number of fixation points, saccade number, and average pupil diameter as experimental indicators for data monitoring. Based on the eye movement heat map obtained by data visualization processing and the results of correlation analysis, we summarized the eye movement behavior characteristics of individuals when viewing the restorative landscape. We try to construct a quantitative evaluation model of the landscape mental recovery benefit with the objective eye movement index as the independent variable through the method of curve estimation. The study results show that individual eye movement behavior is related to the landscape type and the level of psychological recovery is also different. (1)The more singular that the constituent elements are, the more widespread and concentrated the regional distribution of individual attention areas, and the relative psychological recovery benefit is relatively weak. The more complex that the constituent elements are, the more scattered and smaller the individual interest area, and the psychological recovery benefit is better. Brightly colored, dynamic landscapes are easier to form areas of interest to improve the psychological response to the human body. (2) The psychological recovery benefit of the landscape is directly proportional to the changing trend of the average blink duration, number of fixation points, and number of saccades and is inversely proportional to the changing trend of the average gaze length. (3) The objective eye movement index of average blink duration can quantitatively predict the psychological recovery benefit value of the landscape environment. The number of fixation points, the number of saccades, and the average fixation length could predict the psychological recovery benefits of the landscape, while the other indicators had no prediction effect.

**Keywords:** urban landscape space; mental health; restorative environment; eye movement technology

## 1. Introduction

With the rapid development of urbanization, the improvement of people's living standards has also brought about many problems such as crowded population, fierce competition, etc. With the increasing pressure of life, people's negative emotions have increased [1]. The rapid operation of the city has brought people a series of physical and psychological problems. Especially in the context of the global epidemic outbreak, people's work and life pressures have surged, resulting in the incidence of depression, anxiety, insomnia, and other psychological disorders and hypertension, heart disease, and other

physiological diseases being on the rise [2–6]. In the increasingly prominent physical and psychological diseases of urban residents, the urban park landscape space, as the main area for leisure, sightseeing, and exercise activities in people's daily life, assumes the function of social and public health service and plays an important role in the field of urban residents to relieve pressure, relieve anxiety, and restore their physical and mental health. Therefore, it is of great significance to study the action mechanism and influencing factors of urban park landscape spaces' recovery benefit, explore the recovery benefit prediction model of landscape environment, and guide the design practice to improve the physical health of urban residents.

Some environments can alleviate this negative state and restore people's physical and mental health. Such environments are known as "restorative environments". The concept was first proposed in 1983 by American scholars Kaplan and Talbot, an environment that can make people better recovers from psychological fatigue and the negative emotions associated with stress [7–9]. In 1989, Kaplan et al. formally proposed the theory of attention recovery, noting that distance away (being away), ductility (extent), charm (fascination), and compatibility (compatibility) are the four qualities of a good recovery environment [10,11].

According to previous studies, we can see that the factors affecting the benefits of landscape restoration are numerous and complex. From the view of landscape itself, the element composition, spatial type, and the color, form, and collocation of the landscape elements will all affect the restorative experience of individuals in the environment. From the view of individuality, people of different age, occupation, economic income, and childhood background will have different landscape preferences; these will also affect the restorative experience of landscape environment. However, although the factors that affect the restorative benefits of the landscape are extensive, its research framework still has some reference cases. In terms of quantitative research on the visual landscape, Tveit et al. presented a transparent and theory-based scheme for analyzing visual character. They proposed nine key visual concepts in their study: "coherence, disturbance, historicity, visual scale, imageability, complexity, naturalness and ephemera". They presented these nine concepts in a framework of four levels of abstraction. They point out that visual quality is a holistic experience of the nine concepts. The framework and theory proposed in their study provide a reference for the quantification of landscape recovery benefits in this study [12]. At present, most of the related studies on psychological recovery of the landscape focus on three aspects:" restorative perception and evaluation, restorative environment and individual preference, and restorative perception and emotional attachment [13–15]. For example, Wu et al. (2021) explored the relationship between design perception intensity, preference, recovery, and eye movement of an urban landscape. The results showed that landscape design intensity has a significant influence on preference and recovery, and there is a significant positive correlation with eye movement indicators [16]. Liu et al. (2020) linked local landscape characteristics and local attachment with the restorative perception of urban park tourists, verifying that local landscape characteristics, local attachment, and identity have a positive effect on restorative perception [17,18]. In recent years, the research of landscape evaluation has focused on beauty evaluation, satisfaction evaluation, quality evaluation, and ecological risk evaluation. There are few studies on the evaluation of landscape restoration, and most of the studies determine the recovery benefits of landscape by scale and scoring based on individual subjective evaluation [11,19–21]. This method has a strong personal subjective consciousness, and the results will be affected by personal factors such as the physiological state and emotional state of the grader. In order to avoid this experimental error that may be caused by individual subjective consciousness, a few studies have also started to explore the psychological recovery benefits of landscape through eye movement technology. Most of its application is to predict the level of psychological restoration brought by the landscape through the changing trend of eye movement indicators. There are few reports on the evaluation model of the psychological recovery benefits constructed through eye movement indicators [22–28].

The 'eye' is called the window of the human mind. People rely largely on the visual information fed back by eyes to perceive the world. Eye movement characteristics can reflect the law of people's psychological activities. Therefore, eye movement tracking technology has been widely used in various fields. Eye-tracking technology began to mature in the late 19th century. Early applications focused on areas such as reading, product testing, and advertising design [29,30]. With the rapid development of computer technology and the continuous improvement of the measurement accuracy of eye movement index, the application scope of eye movement technology has been greatly expanded. It has gradually been introduced into the research fields of language acquisition, multimedia application, design psychology, geography, cartography, and traffic safety [31–37].

In the 1990s, European scholars introduced eye movement tracking technology as an indicator of quantification method into the field of landscape research for the first time [38,39]. In recent years, this technology has been widely used and shows a growing trend in domestic and foreign landscape fields. The current eye movement research of the landscape mainly focuses on the fields of aesthetic, practical, and curative effects, quantitative evaluation, and analysis from the perspective of landscape vision [40–44]. Among them, in the early study of landscape efficacy, a questionnaire and scale are often used as the method to quantify the restorative benefits of the landscape; however, with the maturity of this technology, people's psychological activities can be reflected by more objective eye movement data, and the technology has begun to be widely used in the study of the landscape recovery effect. For example, eye movement technology is used to explore the difference between the restoration effect of the urban landscape and green landscape and used to compare the difference of eye movement characteristics when facing different degrees of restoration landscape [45–53]. Currently, eye-tracking technology has been extended to various fields and applications, such as memory, classification, sequence learning, face recognition, action recognition, object recognition, and social cognition. In addition, eye tracking technology is also widely used in experimental tasks such as reading, scene perception, and visual search and is often used to study attentional effects in oral comprehension. The main studies associated with the restorative benefits of the landscape include the following. Lin et al. [54] compared and analyzed the eye movement data of different landscape elements through eye movement experimental research, so as to obtain the types of landscape elements with restorative benefits. Sun et al. [55] explored the visual perception of landscape majors and non-landscape majors by comparing the effects of landscape photos with different attributes and showing different characteristics on individual perception. Compared with traditional behavioral measurement methods, eye-tracking technology has the advantages of high ecological validity, high temporal accuracy, spatial resolution, and rich data in spatial, both temporal and physiological, dimensions. In the above-related study of eye movement tracking technology, this study mainly draws on the eye movement experimental techniques commonly used in the field of medicine and psychology. Through the measurement of eye movement indicators, a connection is established between the objective landscape data and the individual subjective restorative psychological data, so as to reflect the influence law of visual elements on the overall restorative benefits of the landscape environment in a more objectively and truly way.

In recent years, researchers in landscape architecture-related fields have been keen to use mature technologies in various interdisciplinary disciplines to conduct related research in the field of landscape architecture. The use of the technical means, research methods, and relevant research experience in various disciplines and different fields has been achieved synthetically to conduct research in landscape architecture field, transform and spread the theoretical results to practical application, and provide practical reference for designers, planners, and city managers. It is now a development trend in landscape architecture-related research direction [56–59]. Based on all the above, this study chose to construct a quantitative function evaluation model of the landscape psychological recovery benefit through the objective physiological eye movement index data. It is expected that the research experience in this case can provide reference ideas for subsequent researchers

and provide data support for the improvement of the recovery benefits of the landscape environment on human mental health.

The content of this study can be summarized as the following points: (1) exploration of the areas of attention on different types of landscapes and extraction of landscape elements that can arouse individual interest in the landscape environment, (2) screening out the types of eye movement indicators which are related to the benefits of landscape recovery and obtaining the eye movement behavior characteristics that individuals show when viewing a landscape and feeling a better recovery effect, (3) exploring the quantitative relationship between the eye movement index and the landscape recovery benefit and establishing a mathematical function model to predict the recovery benefit of the landscape environment, and (4) putting forward suggestions on the restoration benefit construction of a landscape environment. Through this study, we expect to understand the psychological restorative benefits brought by various types of landscapes in urban parks and the performance of such restorative benefits in individual eye movement behavior.

## 2. Materials and Methods

### 2.1. Study Area

In this study, we selected Xihu Park, Zuohai Park, Nanjiangbin Flower Sea Park, Minjiang Park, Jinji Mountain Park, and Fu Road Park as study areas. A schematic representation of the study area is shown in Figure 1 which includes three kinds of urban landscape spaces: comprehensive urban parks, special parks, and greenways. The above areas are all the landscape environment with a long construction year and a high degree of management in the region. Through the chosen site and photos taken in the areas mentioned above, the landscape pictures we obtained can represent the environment of Fuzhou to a certain extent, and the experimental results can reflect well the restorative effect of the landscape environment of Fuzhou on human mental health. Detailed introductions of the study areas are shown in Table 1.

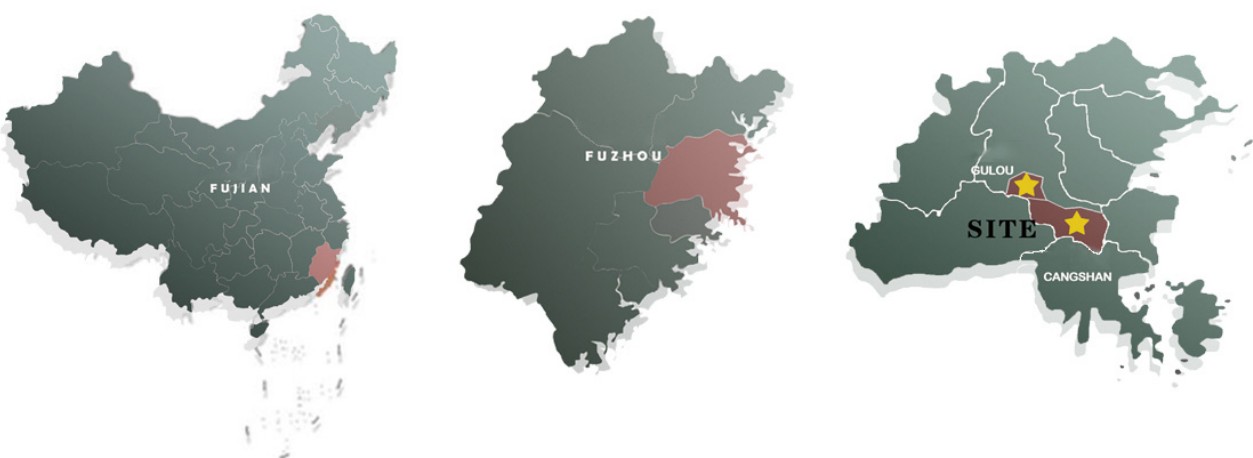

**Figure 1.** Distribution map of the study area.

### 2.2. Participants

In the pre-experiment, in order to ensure that the ranking of the psychological recovery benefits of landscape pictures we obtained was accurate, the experts who we chose to participate in the pre-experiment have all been engaged in relevant work or study in the field of landscape architecture for more than 5 years. According to their rich research experience, they can make more objective and accurate judgments. A total of 100 participants participated in the formal experiment. In order to ensure that the subjects can understand and implement the experimental process accurately as required, we selected the students with a learning background in landscape architecture-related majors, and in order to improve the applicability of the experimental results, the subjects we selected

include three educational levels: senior undergraduates, master's students, and doctoral students. All the participants are between 20 and 30 years old. The experimental results obtained from this group are more accurate and representative. Since the experiment involved eye movement, the participants were also required to have naked eye vision or corrected vision above 1.0 and were required to have normal color vision in both eyes. We selected students with landscape architecture background as subjects to ensure that the subjects understood the experimental process and experimental language as well as the feedback of the experimental results more accurately. In addition, choosing a subject group with three educational levels is more representative than a subject group with a single educational level.

**Table 1.** Characteristics of the study area.

| Green Space Name | Proportion | Features |
|---|---|---|
| Xihu Park | 42.50 hm$^2$ | Located in Gulou District, Fuzhou city, it was built in 1914 and is a comprehensive urban park. It is the most complete classical garden in Fuzhou, with many historical and cultural sites preserved. |
| Zuohai Park | 35.47 hm$^2$ | Located in Gulou District, Fuzhou city, it was first built in 1990 and is a comprehensive urban park. The overall design is "Five continents scenery" as the theme, and the Japanese garden reflects the characteristics of the Japanese courtyard. |
| Nanjiangbin Flower Sea Park | 27.40 hm$^2$ | Located in South Jiangbin Avenue, Fuzhou city, it was first built in 2013 and is a special park. It is famous for its super-large flower sea, integrating leisure, viewing, ecology, and fitness in one. |
| Minjiang Park | 74.01 hm$^2$ | Located in Jiangjiangxi Avenue, Fuzhou city, it was first built in 2000 and is a comprehensive urban park. The north garden has the unique cultural characteristics of the Minjiang River basin, and the south garden has relatively few traces of artificial carving. |
| Jinji Mountain Park | 110.00 hm$^2$ | Located at the foot of Jinji Mountain in Jin'an District, Fuzhou, it was first built in 1997 and is a comprehensive urban park. There are many places of interest in the park, beautiful natural scenery, and strong cultural heritage. |
| Fu Road Park | Total loop length: 19 km | Fu Road is arranged along the ridge line of Jinniu Mountain, connecting the Zuohai plank road around the lake in the northeast, connecting the Minjiang River in the southwest, and running through the five parks in Fuzhou, connecting more than a dozen natural and cultural landscapes. |

Table source: The author arranges and drew according to "Fuzhou Urban Green Space System Planning (2015–2020)" and network related data.

*2.3. Stimuli*

All the landscape images used in the experimental species were taken in the field we mentioned above. In order to avoid the large number of visitors that may increase the difficulty of shooting and the inconsistent quality of landscape photos caused by different light levels which may make for experiment error, we choose to start the research on a sunny and bright non-weekend day. We shot 20 views in each green space area that we selected. The ratio of length to width of the photos is controlled as 4:3. Finally, 120 landscape images were taken in the 6 selected study areas in total. In order to ensure the photo is taken from a human view and to avoid a large number of figures and tall gray buildings appearing in the picture., the shooting point we chose is about 1.6 m away from the ground, and we control for the sky to account for about 1/3 of the whole picture. After obtaining the photo, the size format was unified to 1024 × 768 px through Photoshop software(PS CS5).

The landscape can be divided into 7 categories according to the different constituent of elements, as shown in Table 2. Since some types of landscape are very uncommon in real views, according to the different combinations of landscape elements included in the

actual photos, the visual landscape of green space in an urban park is divided into four categories: green landscape, blue and green landscape, green and gray landscape, and blue, green, and gray landscape. Among the 120 images obtained from the survey, four of each type of landscape photos were selected according to the principle of randomization, and all of them were divided into four groups. Each group contains four types of landscape photos, which were used as the picture materials for the pre-experiment (show as Table 3). The A–D landscapes shown in Figure 1 were used as the picture materials for the formal experiment (Figure 2).

**Table 2.** Classification of landscape element types.

| Landscape Type | Green Landscape | Blue Landscape | Gray Landscape | Blue and Green Landscape | Gray and Green Landscape | Gray and Blue Landscape | Blue, Green, and Gray Landscape |
|---|---|---|---|---|---|---|---|
| Elements constitute | green landscape elements | blue landscape elements | gray landscape elements | green and blue landscape elements | green and gray landscape elements | blue and gray landscape elements | green, blue, and gray landscape elements |

Note: Green landscape elements include trees, shrubs, grass, flowers, and other plant landscape. Blue landscape elements include natural water bodies, artificial lakes, waterscape pools, fountains, and other water landscapes. Gray landscape elements include pavement, landscape structures, landscape sketches, park infrastructure, etc.

**Table 3.** Pre-experimental stimulation material.

| Group | Green Landscape | Gray and Green Landscape | Blue and Green Landscape | Blue, Green and Gray Landscape |
|---|---|---|---|---|
| 1 | | | | |
| 2 | | | | |
| 3 | | | | |
| 4 | | | | |

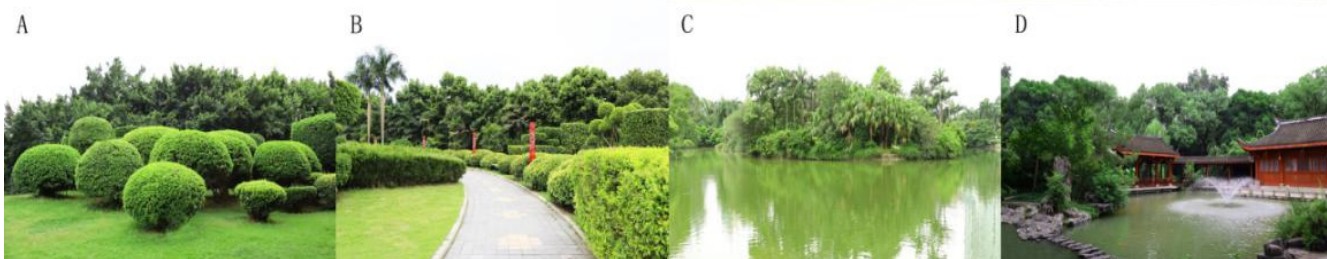

**Figure 2.** Formal experimental stimulation material ((**A**): Green Landscape; (**B**): Green and Gray Landscape; (**C**): Green and Blue Landscape; (**D**): Green, Blue and Gray Landscape).

*2.4. Measurement Tools*

Eye movement instrument: The eye tracker used in the study is the model Eye link 1000 plus manufactured by SR research company, with a sampling frequency of 1000 Hz. The test computer is a Dell P1941S with a screen size of 19 inches. This study draws on the experience of previous research, through eye tracking technology, and selects the average eye duration, average gaze length, average saccade amplitude, blink number, gaze number, saccade, and average pupil diameter. With these seven eye movement indices to conduct data monitoring, we try to explore the individual eye movement behavior in a recovery landscape environment and try to build a quantitative function relationship model between objective eye movement data and subjective evaluation data for landscape recovery benefit.

Self-rated recovery scale: The subjective assessment value of the psychological recovery benefits of the landscape environment in this study is obtained through the self-rated recovery scale (SRRS), which includes four dimensions of the recovery environment: being away, extent, fascination, and compatibility. A total of 22 questions were included in the scale. It is an authoritative and widely recognized measurement scale used by researchers in the field of landscape environment and human mental health research.

*2.5. Experimental Design*

Research contains a pre-experiment and a formal experiment. We exclude all the photos that were wrong or have too many people in the picture from all the photos taken and select all the pictures that meet the basic requirements of the experiment. Then, all the photographs were classified and divided into four categories of images required by the experimental independent variables (green landscape, gray and green landscape, blue and green landscape, and blue, green, and gray landscape). According to the randomization principle in psychological experimental research method, we randomly selected four images in each category of photographs as experimental materials for the pre-experiment. The pre-experiment required 10 experts to sort the recovery benefit of 16 landscape pictures according to the self-assessment recovery scale. It should be particularly noted that the above-mentioned subjects refer to the group of experts with more than 5 years of study background in landscape architecture. They can accurately grasp the purpose of the experimental research and give professional and accurate feedback. According to the results, we chose the pictures which have optimal recovery benefit in the four types of landscape pictures and make these pictures the representative pictures of the four types of landscape for the formal experiment. The formal experiment included 100 participants. The contents can be summarized as two points: (1) measure the eye movement index data of individuals when viewing different types of landscapes and (2) evaluate the subjective recovery benefits of individuals on the viewing landscape environment. In order to avoid the interference brought by the complex environmental factors of the outdoor space as well as the personal factors, this experiment is a double-blind experiment in an indoor laboratory space. Before the beginning of the experiment, the subjects were trained, including the introduction of the instrument, the experimental process, and the interpretation of the experimental questionnaire questions and the relevant professional terms of the options. The purpose is to enable each subject to obtain accurate eye movement data during the

process of the experiment and reflect their own psychological feelings truly and accurately in the process of filling out the questionnaire.

### 2.6. Experimental Procedure

The formal experiment mainly includes two stages: eye movement index measurement and recovery evaluation. The specific experimental process is shown in Figure 3. The objective eye movement measurement experiment was first conducted to obtain the objective eye movement index data. After the participant sat down as required, we need to adjust the eye monitor. Then, after it was confirmed that the subject understood and had no questions, the nine-point pupil correction was started through the computer screen. The experimental stage of pressure application started after the calibration was completed. The specific process is to require the subject to reverse the number breadth; that is, the main test reports a string of numbers, and the subject needs to retell the corresponding number string in reverse sequence, For example, the subject test gives the number string "1, 2, 3", and the subject needs to repeat it as "3, 2, 1". If the subject retells the error number string, he needs to repeat it again. If it is retold correctly, the main test will increases a number to continue the process. The entire pressure application process lasted 60 s. The purpose of this step is to raise the attention and stress of the participant to a certain level so that their performance in the subsequent experimental stage can better reflect the recovery level of the relevant stimulation materials. After the pressure application phase, the experimental photos were automatically presented as a single slide in a random playback order, with each photo played for 30 s. Each photo will be applied to the pressure before playing, and after each photo is played, an offset correction will be performed to reduce the experimental error and ensure the accuracy of the experimental data. Until the four groups of eye movement data are collected, the test stage of eye movement data collection is considered to be finished. After the end of the eye movement experiment, the second stage of subjective recovery feeling assessment was conducted, and the subjects scored the 22 items of the self-assessment recovery scale according to their recovery feeling, with the score range ranging from −3 to 3 points. Each subject was repeated until all the 100 subjects were completed; then, the experiment ended.

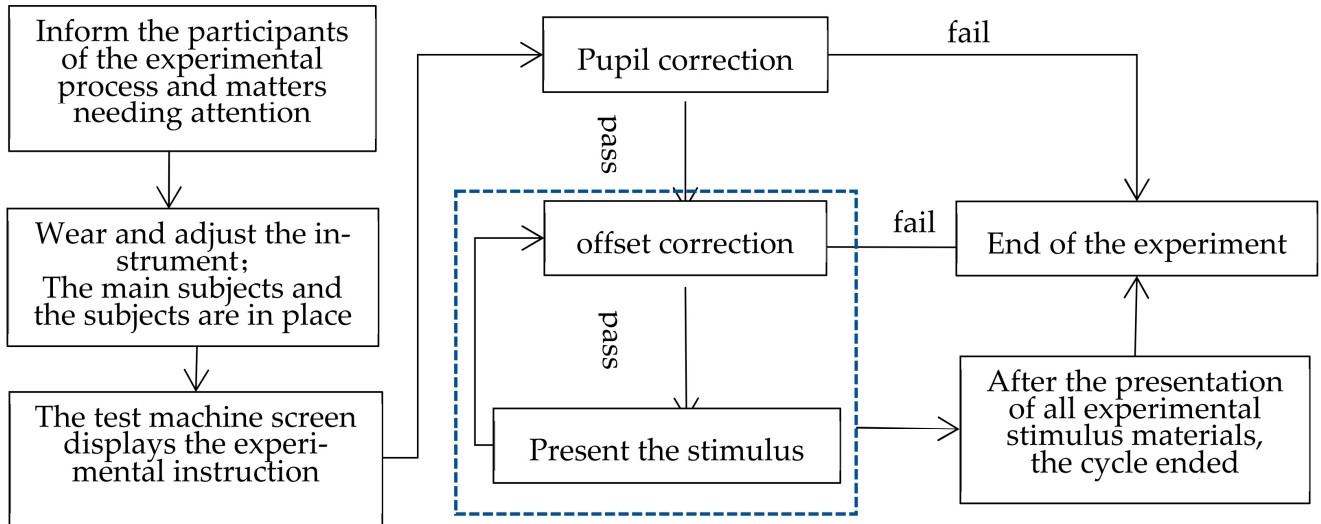

**Figure 3.** Visual experimental flow diagram.

### 2.7. Statistical Analysis

After obtaining the data through experiments, three methods of eye movement analysis (mathematical statistical analysis, qualitative analysis, and quantitative analysis) were mainly used for data processing and analysis. Among them, the regulations and change trends of eye movement index data were analyzed and compared under different experi-

mental conditions to find out the relationship between the date of eye movement and the subjective recovery score. Through mathematical statistical analysis, we explored the characteristics such as the correlations between the basic mathematical variables of the sample such as the mean, variance, etc. Through qualitative and quantitative analysis, a functional relationship is established between the objective data of landscape eye movement index and the subjective recovery feeling evaluation value data, and a functional model that can transform the objective quantitative data into people's subjective psychological perception amount was constructed.

In this study, the eye-movement data were first visualized through the date view platform to obtain the eye movement heat map; then, the data were visualized through analysis and compared to explore the areas of concern for individuals when viewing different types of landscapes to find out the elements that can arouse individual interest. Secondly, through the Pearson correlation analysis, the types of eye movement indicators that were significantly related to the landscape recovery benefit and the eye movement behavior characteristics related to the recovery were selected. Thirdly, through the method of curve estimation, the linear function, logarithmic function, inverse function, quadratic function, and cubic function were used as the fitting model for parameter estimation and fitting degree analysis to obtain the optimal correlation function model between the eye movement index and the landscape recovery benefit.

## 3. Results

### 3.1. Visualization Results of the Eye Movement Data

The eye movement data were visualized, and the eye movement heat map is shown in Table 4. The gradient of the color block from green to yellow to red indicates that the average fixation duration of the subject to the area covered by the color increases during the experiment. It is very intuitive to see that individuals have obvious differences in the areas of interest when viewing different landscape environments.

**Table 4.** Summary of the thermal maps of the area of interest.

| | Green Landscape | Gray and Green Landscape | Blue and Green Landscape | Blue, Green, and Gray Landscape |
|---|---|---|---|---|
| thermal maps | 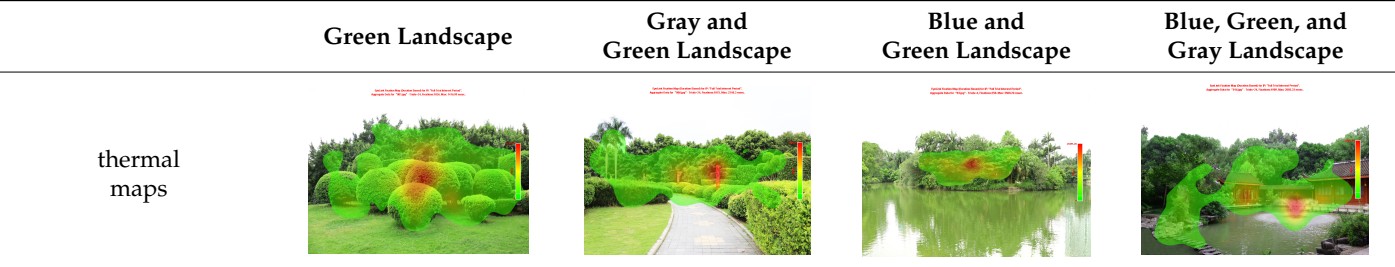 | | | |

According to the eye movement heat map (Table 4) obtained after the visualization processing, it can be seen that individuals have obvious differences in their interest areas when viewing different landscape environments, and their differences are manifested in the regional range of attention areas, distribution regulations, elements, etc. The summary is shown in Table 5. It can be seen that (1) in the green landscape environment space, the attention area of individuals is the most extensive among the four types of landscape environment. The interest area is relatively evenly distributed in the front, middle, and back landscape areas in the center of the picture, and the interest points are concentrated in the artificial plant landscape in the middle landscape area. (2) In the gray–green landscape space, the attention area of the subjects is concentrated in the middle and back areas of the picture, and the most noted elements are in the green landscape area, etc. The interest points are obviously concentrated in the red landscape lamp column in the picture and the end of the road. (3) In the blue and green landscape environment, the attention area is the most concentrated among the four types of landscape space. The sight of the subjects is significantly gathered in the green plant element area at the back of the picture, while

no attention is paid to the prediction of the waterscape area. (4) In the blue, green, and gray landscape space, the distribution of the subject interest areas is the most scattered among the four types of landscapes. The most concerning area is the artificial fountain landscape in the landscape area in the picture and the viewing pavilion in the center of the picture, followed by a landscape building on the right of the scene area and the water step on the left of the foreground area. Comparing this result with the blue and green landscape, it can be seen that individuals pay no special attention to the quiet waterscape in the landscape environment, while the dynamic waterscape will obviously significantly enhance the attention of individuals.

**Table 5.** Summary table of eye movement area of interest analysis.

| Title 1 | Green Landscape | Gray and Green Landscape | Blue and Green Landscape | Blue, Green, and Gray Landscape |
|---|---|---|---|---|
| Landscape composition elements complexity | Level 1 | Level 2 | Level 2 | Level 3 |
| Area of interest | Level 4 | Level 2 | Level 1 | Level 3 |
| regularities of distribution | Centralized distribution of areas of interest Centralized distribution of interest elements | Centralized distribution of areas of interest Distal distribution of elements of interest | Centralized distribution of areas of interest Centralized distribution of interest elements | Distal distribution of regions of interest Distal distribution of elements of interest |
| Interest elements | Focus on the artificial modeling plant landscape in the landscape area, with no significant element characteristics | The dispersion is concentrated in the brightly colored structures and at the end of the road | Focus on the plant landscape elements in the picture area, with no significant element features | Spread is concentrated in dynamic water features, water steps, and colorful buildings |

*3.2. Correlation Analysis between Eye Movement Index and Landscape Recovery Benefit*

A total of 400 landscape recovery benefit score data points were obtained through the experiment, and the eye movement performance characteristics of the visual landscape mental recovery were explored by conducting the correlation analysis between the seven selected eye movement index data and the mental recovery score data. ("Indicator 1–Indicator 7" in Table 6 represents mean blink duration, mean gaze length, mean saccade amplitude, number of blinking, number of gaze points, saccade number, and mean pupil diameter, respectively). According to the results shown in Table 6, it can be seen that for the three eye movements (mean saccade amplitude, blink number, and average pupil diameter), there are no significant correlation between them and the landscape recovery benefits. Among the remaining four eye movement indicators which have a correlation with the landscape recovery benefits, the correlation rank between average blink duration, average gaze duration, and landscape recovery benefit was extremely significant. The correlation rank between the number of fixation points, the number of saccades, and the landscape recovery benefit was significant. In addition, except for the correlation between the average fixation length and the landscape recovery benefit being negative, the other three indicators of the eye movement all showed a positive correlation with the landscape recovery benefit.

**Table 6.** Results of correlation analysis between landscape restoration and eye movement index.

|  |  | Metric 1 | Metric 2 | Metric 3 | Metric 4 | Metric 5 | Metric 6 | Metric 7 |
|---|---|---|---|---|---|---|---|---|
| Recovery grade | pearson correlation | 0.133 ** | −0.366 ** | −0.051 | 0.038 | 0.117 * | 0.120 * | −0.024 |
|  | significance (double-tail) | 0.008 | <0.001 | 0.305 | 0.444 | 0.019 | 0.016 | 0.638 |
|  | the number of cases | 400 | 400 | 400 | 400 | 400 | 400 | 400 |

Note: * indicates a significant correlation between the independent variable and the dependent variable. ** indicates an extremely significant correlation between the independent variable and the dependent variable.

### 3.3. Quantitative Evaluation Model of Landscape Resilience

Among the correlation coefficients of the four eye-movement indicators which have significant correlations with recovery benefits, the largest absolute value is 0.366. It can be seen that the linear relationship between these four indicators and landscape recovery benefits is not very obvious; therefore, we use the curve estimation method and choose a linear function, logarithmic function, inverse function, quadratic function, and cubic function as five functions for a fitting model to explore the function between the four eye indicators and landscape recovery benefits. The fitting results of eye movement data for average blink duration, average gaze duration, number of fixation points, saccade times, and landscape recovery data are shown in Table 7 and Figure 4.

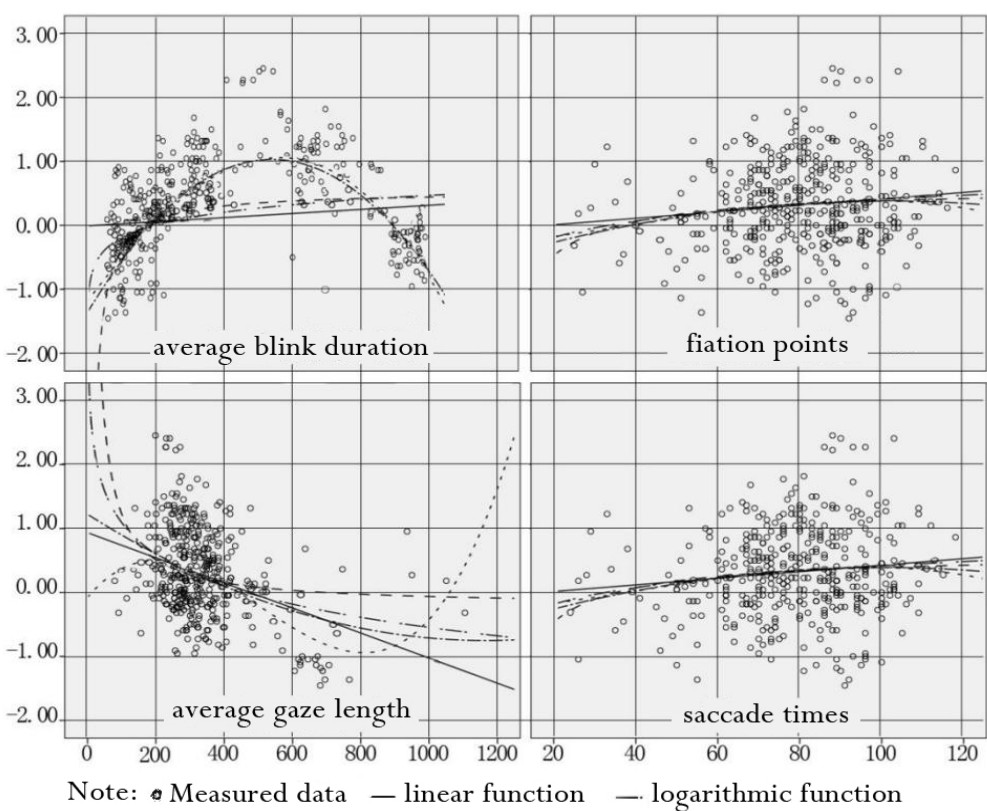

**Figure 4.** Fitting function of eye movement index and landscape restoration score.

**Table 7.** Results of parameter estimation of eye movement model of restoration.

| Variable | Model Summary | | | | | | Parameter Estimates | | | |
|---|---|---|---|---|---|---|---|---|---|---|
| | Function | $R^2$ | F | df1 | df2 | Sig | Constant | b1 | b2 | b3 |
| Recovery benefit score and average blink duration index | linear | 0.018 | 7.157 | 1 | 398 | 0.008 | 0.184 | <0.001 | / | / |
| | logarithmic | 0.097 | 42.558 | 1 | 398 | <0.001 | −1.273 | 0.277 | / | / |
| | inverse | 0.175 | 84.478 | 1 | 398 | <0.001 | 0.706 | −88.958 | / | / |
| | quadratic | 0.549 | 241.271 | 2 | 397 | <0.001 | −1.128 | 0.009 | $-7.926 \times 10^{-6}$ | / |
| | cubic | 0.552 | 162.630 | 3 | 396 | <0.001 | −0.956 | 0.007 | $-3.928 \times 10^{-6}$ | $-2.567 \times 10^{-9}$ |
| Recovery benefit score and average gaze length index | linear | 0.134 | 61.582 | 1 | 398 | <0.001 | 0.956 | −0.002 | / | / |
| | logarithmic | 0.117 | 52.697 | 1 | 398 | <0.001 | 4.347 | −0.701 | / | / |
| | inverse | 0.061 | 25.927 | 1 | 398 | <0.001 | −0.166 | 141.830 | / | / |
| | quadratic | 0.140 | 32.313 | 2 | 397 | <0.001 | 1.239 | −0.003 | $1.414 \times 10^{-6}$ | / |
| | cubic | 0.177 | 28.297 | 3 | 396 | <0.001 | −0.039 | 0.006 | $-1.901 \times 10^{-5}$ | $1.249 \times 10^{-8}$ |
| Recovery benefit score and gaze point number index | linear | 0.014 | 5.565 | 1 | 398 | 0.019 | −0.092 | 0.005 | / | / |
| | logarithmic | 0.016 | 6.280 | 1 | 398 | 0.013 | −1.291 | 0.368 | / | / |
| | inverse | 0.015 | 6.091 | 1 | 398 | 0.014 | 0.593 | −21.322 | / | / |
| | quadratic | 0.017 | 3.462 | 2 | 397 | 0.032 | −0.626 | 0.020 | $-9.899 \times 10^{-5}$ | / |
| | cubic | 0.017 | 2.335 | 3 | 396 | 0.073 | −0.303 | 0.004 | 0.000 | $-1.065 \times 10^{-6}$ |
| Recovery benefit score and saccade number index | linear | 0.014 | 5.803 | 1 | 398 | 0.016 | −0.096 | 0.005 | / | / |
| | logarithmic | 0.016 | 6.554 | 1 | 398 | 0.011 | −1.299 | 0.370 | / | / |
| | inverse | 0.016 | 6.358 | 1 | 398 | 0.012 | 0.594 | −21.141 | / | / |
| | quadratic | 0.018 | 3.605 | 2 | 397 | 0.028 | −0.630 | 0.020 | <0.001 | / |
| | cubic | 0.018 | 2.430 | 3 | 396 | 0.065 | −0.321 | 0.005 | <0.001 | $-1.054 \times 10^{-6}$ |

According to the function results of the four indicators shown in Figure 4, the data points of the average blink duration index are more consistent with the fitting function curve, while the data points of the number of fixation points, the average gaze length, and the saccade times are scattered and the regularity is not obvious.

According to the parameter estimation results of the landscape restoration eye movement index model (Table 5), it can be seen that among the five fitted models of the average fixation duration and the landscape restoration benefit, the cubic function has the largest determination coefficient of 0.177. The degree of fit of the model is very low. The prediction confidence of these five types of functions is not high, and there is no significant regularity between them. Therefore, we can only infer that there is a very significant correlation between them ($-0.366$), which are to say that for the mean, the average fixation length of the landscape is shorter when the recovery benefit is higher, and the relationship cannot be described by quantitative function expression. Similarly, in the model of the number of fixation points, the number of saccades, and the landscape recovery benefit, the fitting degree of the five types of function models is less than 0.1, indicating that there is also no obvious mathematical function relationship between the two eye movement indicators and the landscape recovery benefit. In the five types of fit models to describe the relationship between eye movement indicators and landscape restorative benefits, the best fitting degree of it is 0.552; therefore, the recovery benefit of the landscape environment can be predicted by the objective eye movement index of the average blink duration. Based on the results of the model parameter estimation, the functional relationship model between them can be expressed as follows: $R = -0.956 + 0.007B - 3.928 \times 10^{-6}B^2 - 2.567 \times 10^{-9}B^3$. R represents the restorative benefit of the landscape, and B represents the mean blink duration.

## 4. Discussion

### 4.1. Individual Attention Areas and Characteristics of Elements for Different Types of Landscapes

To sum up, we find that: (1) A single landscape environment and a high complex landscape environment will attract a wider area of interest than a medium complex landscape, and with the increase of landscape composition element complexity, the individual attention area in the ornamental landscape will become more scattered. This may be due to the fact that as the higher complexity of the landscape environment visitors interested in landscape element probability is relatively high, visitors' interest area with the distribution of interest elements also becomes more scattered. (2) The road edge line has a line-of-sight guidance effect, which will attract the individual's attention to the end of the road extension line. (3) Individuals are more likely to pay attention to dynamic, brightly colored, or landscape elements with special artificial shapes. Landscape color and state will cause individual eye movement behavior. The bright colors, special shapes, and dynamic landscape are more likely to make individuals have an interest in further exploration.

### 4.2. Characteristics of Landscape Mental Recovery

According to the correlation analysis of eye movement indicators and landscape recovery benefit, it can be seen that the recovery benefits brought by a landscape can be feedback for four eye movement indicators: average blink duration, average fixation length, number of fixation points, and number of saccades. When the landscape environment brings people a good restorative experience, the reasons for the objective physiological behavior feedback by individuals and the effect of affecting the individual restorative experience are analyzed as follows: (1) In the process of viewing the landscape, the frequency of blinking decreases, and the speed slows down, which alleviates the stress and tension. (2) The average duration of all gaze behaviors generated in the viewing process is short, which indicates that individuals receive good information in the landscape picture, and the difficulty of information processing is low. The landscape information in the picture does not need individuals to spend too much attention to interpret, so there is not much fatigue. (3) In the process of viewing the landscape picture, there are more fixation points, which indicates that individuals have more interested areas in the landscape shown by the picture, and the

landscape has a strong attraction to them. This result is also in line with the characteristics that Kaplan proposed that the restorative environment should be attractive. (4) Saccades are produced in the behavior of a visual landscape, meaning visual information search behavior, which reflects that the landscape picture does not cause strong special stimuli to individuals, and individuals have no resistance to the landscape environment, which is conducive to the relaxation of nerves.

*4.3. Quantitative Evaluation Model of Landscape Restoration and Its Application*

The landscape recovery benefit evaluation model obtained in this study is based on the instrument measurement of eye movement data to measure the visitors in the real landscape environment. Its advantage is that the instrument can be the most real and accurately reflect the behavior characteristics of people in the landscape environment space, and then the function model was the one that we constructed in this study, Compared with visitors' subjective evaluation of recovery according to their own feelings, this method is more scientific and objective, It will not be affected by individual subjective psychological factors and can provide feedback on the restorative experience benefits obtained by individuals more accurately.

According to the research results, there is a good mathematical function relationship between the average blink duration and the good recovery effect of the landscape. The application of determining the recovery effect of the landscape environment by studying the model can be divided into the following two categories: (1) For the landscape environment which was already built down, by determining the average blink duration data of visitors when viewing different landscape photos or the field environment, we compare the differences between the restoration effects of different landscape environments and then scientifically determine which landscape will need to improve the restoration benefits and how to transform them. (2) For the landscape scheme that is not completed or is still under the design stage, this model can provide data support for the restoration design of the landscape. The eye motion index data of the viewer are measured by designing the virtual model of the model to compare the optimal design scheme.

*4.4. Design Suggestions for Landscape Restoration Benefit Improvement*

According to the analysis of the above parts, it can be learned that the relatively simple landscape environment with the constituent elements makes it easier for people to feel relaxed. The relatively complex landscape environment will make the viewers have more interest points, but it is also easier to consume individuals' directional attention, which is not conducive to relieving pressure and relaxing tension. What should be noted about the above conclusions is that this is an internal comparison of the same type of landscape, and the relatively simple landscape environment usually has a high recovery benefit. This study did not contrast the different types of landscapes and failed to draw a conclusion on the relationship between the complexity of the different types of landscapes and their restorative benefits. But previous studies have found that higher landscape heterogeneity leads to higher perceived restoration by visitors. Therefore, this conclusion needs further experimental verification. Based only on the conclusions drawn from this study, we propose that, when creating or improving the restoration benefits of the landscape environment, we should pay attention to the following points: (1) Control the complexity of the constituent elements of the landscape environment. First of all, we should pay attention to grasping the elements of the environment; that is, they cannot be too single, so that the viewer will have boring feelings, but also not too complicated, resulting in the transition consumption of visitors' directional attention and making them have irritability and anxiety. Attractive elements can be arranged in the overall harmonious landscape environment to attract interest and create feelings of happiness. (2) Pay attention to the color selection of landscape elements and the presentation of the state in the environment. The type and form of the elements in the environment will also affect the overall recovery experience, such that the viewer line of sight will be guided by the road edge line. Its attention is on more

colorful scenery which is attracted by a dynamic landscape environment. The appropriate layout of such elements in the landscape environment helps to improve the landscape's overall recovery benefits. (3) Smart use of linear landscape to guide visitors' line of sight and design restorative landscape elements in the position of landscape line of sight. In the process of restorative landscape design and construction, we should pay attention to the processing of the linear landscape in the environment, such as the route and plant contour landscape skyline. Using these linear landscapes will guide the line of sight to a relaxed, pleasant landscape environment. To effectively attract the attention of the viewer landscape area, we should focus on design.

## 5. Conclusions

This study explored the benefits of psychological recovery and its influencing factors on the landscape environment through an objective eye movement experiment and subjective recovery questionnaire. Through the visual processing of data, correlation analysis, and function fitting, we found that the greater the number of landscape composition elements, the more scattered the individual concerns. The landscape elements that can arouse individual interest in the landscape environment are usually characterized by bright color, unique shape, and a dynamic state. At the same time, we found that a linear landscape in the environment has an obvious guiding effect on the line of sight. Through the fitting and analysis of eye movement index data and psychological recovery questionnaire results, we can know that the four eye movement indicators (average blink duration, number of fixation points, average fixation length, and number of saccades) can be used to predict the recovery benefits that the landscape environment can bring to individuals. The shorter the average fixation time, the longer the duration of the average eye blink, the more the number of gaze points and saccades, and the better the recovery benefits brought by the environment. In addition to the four eye movement indicators concerned in this study, some researchers proposed that there is a correlation between eye blink number, pupil diameter indicators, and the benefit of landscape recovery, and whether there is a functional relationship needs to be further explored. Among the four indicators of concern in this study, there is no regular functional relationship between the number of fixation points, average fixation length, saccades, and the benefit of landscape recovery, and it can only be used as a trend predictor. However, there is a cubic function between the average blink duration and the recovery benefit of the landscape environment. The resulting function relationship can be used to determine the recovery benefit of the built landscape environment to provide corresponding data support for the landscape improvement and then propose a scientific transformation scheme or used to predict the recovery benefit value of the unbuilt landscape design scheme and compare the optimal design scheme.

Based on the above conclusions drawn in this study, we can learn about the different types of landscape characteristics in urban parks, as well as the eye movement behavior characteristics and restorative experience of individuals in various types of landscape space, so that we can improve the restorative benefits of various types of landscape space in urban parks through design means. The details are as follows.

For the green landscape space, we know through our research that its landscape characteristics are that the landscape constituent elements are relatively simple. Participants had a shorter average fixation duration and more saccades when viewing the most obvious eye-movement behavior features of such landscapes. This indicates that subjects are in a more relaxed state in an environment with relatively constituent elements. There are not too many elements in the environment that allow individuals to generate directed attention. For this kind of landscape space, the landscape elements can be appropriately added to the environment. It enables certain elements of the environment to increase the subjects' interest in the viewing process, Eye movement behavior was represented by an increased number of saccades. According to the model derived in our trial, this increase in saccadic behavior is exactly the concrete manifestation of enhancing landscape resilience.

For the gray–green landscape space, it contains moderate types of landscape elements, which are neither too single nor too complex. According to the experimental results, we can see that when the subjects view such landscapes, the fixation points are usually more scattered. Primary attention will focus on individual characteristic landscape elements. Eye movement behavior showed a greater number of fixation points, more saccades, and a shorter average fixation length, which indicated that subjects showed more distracted attention when viewing the gray–green landscape space, and the consumption of non-directed attention will not produce a more directed attention-consuming search inquiry behavior. For this type of landscape space, we can modify some of the elements while ensuring that the constituent elements are moderate. We enhance the attraction of the landscape from the aspects of color and shape so that the interest of the subjects increases and then improves their restorative experience in this kind of landscape space.

For the blue–green landscape space, its characteristics are very similar to the gray–green landscape space, and the composition elements are relatively moderate. However, it is worth noting that, compared with the gray landscape elements, the blue landscape elements are more likely to catch the subjects' love and make them feel a pleasant mood. The dynamic waterscape elements are especially good at attracting the attention of subjects, making them act with more eye movement behavior. In order to improve the restoration benefits of this kind of landscape environment, it is obviously necessary to improve the water elements. Subjects generally have generated sufficient fixation points when appreciating such landscapes. What we need to achieve is enabling subjects to produce more saccadic behaviors to enhance landscape resilience. Obviously, adding the dynamic waterscape elements appropriately is the best option.

For the blue–green–gray landscape space, it has the most complex element among several types of landscape. There is no doubt that rich landscape elements are an advantage of this type of landscape space. However, it is also easy to consume the subject with too much attention and cause a sense of fatigue. Subjects often produce more saccadic and fixation behaviors when viewing such landscape environments. Due to the large number of interest points and the scattered distribution, the mean fixation duration was also relatively short. These all have a positive promotion effect on the landscape restoration benefit. However, participants may be too excited in an excessively rich environment, which is not conducive to relieving fatigue. Determining how to maintain a long blink duration while viewing this landscape is the key to improving their restorative experience. Reducing the number of elements in the environment appropriately and controlling the color and shape of the elements are a good solution.

Landscape restoration depends on multiple factors that must be taken into account. Besides the landscape types and their elemental composition mentioned above, there are many other factors that will influence the restorative benefits of the landscape. In this study, we only explored the eye movement behavior characteristics of subjects in different types of landscape spaces of urban parks. Based on this, we then selected indicators and constructed the model. We expected that subsequent researchers will be able to further explore the influence of many other factors on the restorative benefits of the environment and the corresponding eye movement behavior characteristics. In conclusion, this study is committed to realizing a sustainable development strategic decision to improve the health sense and happiness of urban residents on the basis of protecting the landscape environment. In order to adapt the city and the landscape to the challenges of the current times, it is necessary to establish the concept of sustainable development of the living environment. Based on the discipline of landscape architecture, the research on the health benefits of an urban green spatial landscape is conducive to improving the sustainability of urban development, meanwhile improving the well-being of human settlements and the quality of the living environment. In general, in order to provide reference research ideas and data support for the construction of the restorative environment in future landscape spaces, this study explores the restorative benefits of the visual landscape in green spaces through a series of research methods such as an eye movement experiment and a restorative

subjective evaluation experiment. We try to let more researchers pay attention to the restoration of the visual landscape in green spaces and, at the same time, concentrate on how to apply the results of theoretical research to practice, transform theoretical research language into design language, and truly realize the sustainable practical strategy of an urban living environment.

**Author Contributions:** Conceptualization, X.F. and Q.H.; methodology, X.F. and Y.Z.; software, X.F. and Y.Z.; validation, X.F. and D.K.; formal analysis, X.F.; investigation, X.F. and Y.Z.; resources, X.F. and Y.Z.; data curation, D.K.; writing—original draft preparation, X.F.; writing—review and editing, M.W. and J.D.; visualization, Q.H.; supervision, M.W.; project administration, J.D. All authors have read and agreed to the published version of the manuscript.

**Funding:** This research was funded by Project of Forest Park Engineering Technology Research Center of State Forestry Administration, grant number PTJH15002. Wuyi Mountain National Park Research Institute special project, grant number KJg20009A. Rural ecological product value realization research project, grant number KKy22044XA.

**Institutional Review Board Statement:** Not applicable.

**Informed Consent Statement:** Informed consent was obtained from all subjects involved in the study.

**Data Availability Statement:** The data are not publicly available due to the ongoing research, and the authors will continue to work with it in the future.

**Acknowledgments:** We thank all the study participants in this study.

**Conflicts of Interest:** The authors declare no conflict of interest.

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
