# Peer review of "Quantitative Model Study of the Psychological Recovery Benefit of Landscape Environment Based on Eye Movement Tracking Technology"

_sustainability, doi:10.3390/su151411250_

Round 1
Reviewer 1 Report
Abstract
Too long. too much of the introduction part in your abstract. Make your abstract shorter. Too long.
Study area
Please provide a map of your study area.
Conclusion
The author need to add the element of so what need to be done to out landscape in the future to have more restorative kind of environment? What's our duty? what's need to be done. What are the changes.
Author Response
Thank you for your letter and for the reviewers’ comments concerning our manuscript entitled “Quantitative model study of the psychological recovery benefit of landscape environment based on eye movement tracking technology” (Manuscript ID: sustainability-2427145). Those comments are all valuable and very helpful for revising and improving our paper. We have studied comments carefully and have made correction which we hope to meet with approval. The main corrections in the paper and the responds to the reviewer’s comments are as attachment.

Reviewer 2 Report
1. In lines 74-75, please pay attention to the capitalization of the first letters of the words.
2. In line 79, it should be "et al."
3. When introducing the research of a specific researcher, only the surname is needed, for example, Wu et al. (2021).
4. In line 88, a reference should be added when introducing the research of a researcher.
5. In lines 128-131, the introduction of relevant studies on exploring landscape restoration effects using eye-tracking techniques can be more detailed, especially regarding important research findings.
6. The content in lines 132-142 appears abrupt. It is recommended to shorten or remove it.
7. The introduction section lacks an review of the research on exploring landscape restoration effects using eye-tracking techniques. This type of research has already been conducted, and the author should provide some detailed introductions to these studies, including the research achievements in this field, as well as any limitations. Based on this, the author can elaborate on the research questions addressed in their study.
8. In section 2.1, please provide the location of the study area.
10. There is redundancy between the experimental design in section 2.5 and sections 2.2 and 2.3. It is recommended to make corresponding adjustments.
11. In Table 6, it is suggested to place the dependent variable directly in the first column of the table.
12. The analysis of eye-tracking heatmaps in section 4.1 is recommended to be moved to section 3.1. The discussion section should focus more on the interpretation of results and the similarities and differences with previous studies.
13. The title of Table 7 should be placed above the content of the table.
14. In section 4.4, the author describes, "According to the analysis of the above parts, it can be learned that the relatively simple landscape environment with the constituent elements is easier for people to feel relaxed." However, the paper does not include a differential analysis of restoration scores for different landscape types. Moreover, previous studies have found that higher landscape heterogeneity leads to higher perceived restoration by visitors, which contradicts the author's viewpoint. Therefore, the validity of this conclusion deserves further discussion.
15. Throughout the discussion section, the author lacks in-depth analysis of the results, especially in comparison with previous studies. There have been numerous studies on the relationship between landscape features and restoration, as well as eye-tracking features and restoration. It is recommended for the author to supplement this part of the discussion.
16. Abbreviations used for the first time in the abstract should include their full forms.
17. The abstract should not include research progress or future research directions. This information can be included in the last subsection of the discussion section.
Author Response

(The authors gave the same response as above.)

Reviewer 3 Report
I think the paper can be published in the present form.
Author Response

(The authors gave the same response as above.)
